# In Vitro Susceptibility Tests in the Context of Antifungal Resistance: Beyond Minimum Inhibitory Concentration in *Candida* spp.

**DOI:** 10.3390/jof9121188

**Published:** 2023-12-12

**Authors:** Iacopo Franconi, Antonella Lupetti

**Affiliations:** 1Department of Translational Research and New Technologies in Medicine and Surgery, University of Pisa, 56126 Pisa, Italy; iacopo.franconi@phd.unipi.it; 2Mycology Unit, Pisa University Hospital, 56126 Pisa, Italy

**Keywords:** antifungal susceptibility testing, *Candida* spp. fungicidal tests, antifungal resistance, minimum fungicidal concentration, time-kill curve analysis, serum fungicidal concentration, antifungal tolerance

## Abstract

Antimicrobial resistance is a matter of rising concern, especially in fungal diseases. Multiple reports all over the world are highlighting a worrisome increase in azole- and echinocandin-resistance among fungal pathogens, especially in *Candida* species, as reported in the recently published fungal pathogens priority list made by WHO. Despite continuous efforts and advances in infection control, development of new antifungal molecules, and research on molecular mechanisms of antifungal resistance made by the scientific community, trends in invasive fungal diseases and associated antifungal resistance are on the rise, hindering therapeutic options and clinical cures. In this context, in vitro susceptibility testing aimed at evaluating minimum inhibitory concentrations, is still a milestone in the management of fungal diseases. However, such testing is not the only type at a microbiologist’s disposal. There are other adjunctive in vitro tests aimed at evaluating fungicidal activity of antifungal molecules and also exploring tolerance to antifungals. This plethora of in vitro tests are still left behind and performed only for research purposes, but their role in the context of invasive fungal diseases associated with antifungal resistance might add resourceful information to the clinical management of patients. The aim of this review was therefore to revise and explore all other in vitro tests that could be potentially implemented in current clinical practice in resistant and difficult-to-treat cases.

## 1. Introduction

Fungal infections are on the rise in recent years, as the population at risk of acquiring such infections is dramatically increasing [1,2]. As highlighted by the recent World Health Organization (WHO) report, health-care associated fungal diseases are not only increasing in the target population but also doomed by the rise in antifungal resistance rates, forcing an expert panel to address such clinical phenomena in the fungal pathogen priority list [3].

In the case of yeast infections, *Candida* species represent the most frequently fungal isolate causing invasive diseases in hospitalized patients, with high mortality rates and detrimental consequences for both patients and healthcare resources [1,4,5,6,7,8]. Alongside this, clinicians are witnessing a worrisome surge in antifungal resistance, especially in the case of invasive aspergillosis and non-*albicans* species and particularly with *Candida auris* and *Candida parapsilosis* [9,10,11,12,13] as well as echinocandin resistance in *Candida glabrata* (currently classified as *Nakaseomyces glabrata*) [13,14,15]. Reports from all over the world are continuously depicting increasing azole-resistance rates and, to a lesser extent, echinocandin resistance and tolerance within such isolates, particularly in Western countries [16,17,18,19,20,21,22,23,24]. According to the definition proposed by Berman and colleagues [25], antifungal tolerance is defined as the ability of a subpopulation of a susceptible isolate to grow over a longer incubation period (extended beyond the time required to define MIC values) under drug concentrations above MIC values, without harboring any known genetic resistance mechanism. Infections caused by *C. parapsilosis* occur in the form of hospital outbreaks, thus substituting the underlying *C. parapsilosis* healthcare-associated fungal niche and persisting within the environment and colonizing patients permanently [10,13,16,26,27]. As clonal outbreaks of *C. parapsilosis* are emerging worldwide [16,17,18,19,20,21,22,23,24], new mutations are frequently described within most of the newly reported outbreaks in addition to the previously known ones. These combined together make the single outbreak a one-of-a-kind microbiological and epidemiological phenomenon worthy of further investigation and intervention [12,28]. 

Antifungal resistance is narrowing the already limited therapeutic options available, hindering the possibility of achieving a clinical cure [13]. In addition, subjects at risk of developing healthcare-associated fungal infections are also at higher risk of acquiring invasive fungal infections beyond candidemia, such as osteomyelitis, prosthetic joint infections, endocarditis, uveitis, deep-seated ocular infections, meningitis, and disseminated hepatosplenic candidiasis [29,30]. Still, more threats to come are related to the consequence of climate change on human–fungal pathogenesis and interaction. As elucidated in the One Health approach by WHO and meticulously described in the studies by Nnadi et al. [31] and Coates and colleagues [32], challenges from newer fungal species adapting to human biology, as well as yet-undiscovered newly associated resistance mechanisms, will arise in the sooner future. Among all, for *Candida* spp. isolates, it has been suggested that human adaptation and infections caused by *C. auris* (which represents the paradigm of multi-drug resistant *Candida* spp.) could be indeed connected to climate change. Based on these assumptions, not only is the population that acquires an invasive fungal infection under climate change pressure increasing, but the number of species along with their new antifungal-resistance traits able to adapt to human biology are also expected to increase [32]. The main driver of antifungal therapy to date is the standardized in vitro antifungal susceptibility test aiming at defining minimum inhibitory concentrations (MIC) [33]. Such data must be interpreted in the light of established clinical breakpoints (CBP) for antifungals, which allow for interpretation of the susceptibility profile, defined similarly to antibiotic susceptibility for bacterial agents. These are important values provided by international organizations according to the Clinical & Laboratory Standard Institute (CLSI) and the European Committee on Antimicrobial Susceptibility Testing (EUCAST) respective standardized in vitro testing methods, which have been validated with clinical correlations and outcomes [34]. Therefore, with the strong evidence reported, microbiologists are allowed to define susceptibility profiles of multiple antimicrobial drugs, guiding the clinician to select the best agent for the patient. CBP was shown to have some limitations, particularly in the context of antifungal susceptibility, with the example of fluconazole-resistance interpretation in *C. albicans* isolates with MIC values above 8 mg/L. In this context, 37% of these resistant isolates actually responded to therapy, achieving a clinical cure [34]. It is important to mention that the “90/60” rule also applies to fungal isolates in general. According to such a rule, in the case of documented full susceptibility to the tested antifungal compound, a response to therapy will be observed in 90% of the cases, whereas this percentage is reduced to 60 in the case of reported resistance in the in vitro test [33,35].

Currently resistance and susceptibility to selected antifungals can be inferred via different methods such as the E-test strip, disk diffusion, and broth microdilution. These techniques have become milestones in detecting antifungal susceptibility, driving drug selection and prescription. However, as observed for bacterial infections, in some rare but critical clinical scenarios, where limited data are available and patients develop difficult-to-treat invasive infections, susceptibility profiles and MIC values obtained from routine antifungal susceptibility tests might not be sufficient to drive antifungal therapy, and the risk of associated clinical resistance and failure is high [36,37,38]. It is important to mention that in such particular clinical pictures as those described above, clinical resistance rather than microbiological resistance represents the scariest and most frequent negative outcome [37]. Indeed, standardization of protocols and results interpretation along with reported evidence of correlation with clinical outcomes, MIC values, and susceptibility profiles are not the only factors accounting for treatment success. Underlying comorbidities, site of infection and clinical picture, severity of the infectious syndrome, and comedications might all play a role in the success of therapy [14]. Therefore, in such selected cases, expanding antifungal susceptibility testing beyond the common MIC determination might be beneficial to guide clinicians in the choice of the proper treatment for the benefit of the patient. In this context, the implementation of fungicidal tests as minimum fungicidal concentration (MFC), time-kill curve (TKC) analyses, and serum fungicidal concentration (SFC) that have long been used in research contexts and pre-clinical studies could be a valuable resource to drive antifungal therapy and to select the most active compound as observed for bacterial counterparts [36,38]. In addition, very little is known about the clinical correlation of such tests with antifungal treatment outcomes, thus highlighting that this is an unmet clinical need and might also be a fervid field of research. 

On the other hand, in the context of newly reported antifungal-resistant clinical outbreaks, predicting the development of acquired resistance to the most active compound as a consequence of selective antifungal pressure is another unmet clinical need for both infection control and antifungal stewardship purposes. Another strictly related subject, the antifungal tolerance, is still not properly explored and investigated, as standardized procedures to detect it and studies on its impact and on the development of resistance and its clinical implications are lacking.

The aim of this study is to revise current in vitro non-molecular phenotypic fungicidal test assays beyond quantification of MIC in yeast pathogens that might add useful information to the management of complicated disseminated fungal infections in the context of emerging antifungal-resistant clinical outbreaks. 

## 2. Minimum Fungicidal Concentration

According to the first definition reported by the National Committee for CLSI in the M26-A document in 1998, one of the methods proposed for testing bactericidal activity of antibiotic compounds was the evaluation of the minimum bactericidal concentration (MBC), also known as minimal lethal concentration (MLC) [39,40,41]. Such a test aims at evaluating the minimal antibiotic concentration required to kill 99.9% of the bacterial suspension incubated for 24 h at 37 °C. In particular, MBC is usually performed after a standardized broth microdilution antibiotic susceptibility test is run to establish MIC values for the examined pathogen [39]. Bacterial suspensions above MIC values obtained from the broth microdilution test are further diluted and plated accordingly on agar solid media in order to estimate the viable bacterial count (CFU/mL) after 24 h incubation. By comparing bacterial counts in all suspensions for each well above MIC values, the lowest concentration at which a reduction of 99.9% of the final bacterial inoculum is observed is the MBC. Pfaller et al. [42] in their study translated the concept of MBC to antifungal susceptibility testing. Briefly, minimum fungicidal concentration (MFC) could be defined as a secondary test performed after a routine broth microdilution antifungal susceptibility test where suspensions of yeasts contained in each well above the MIC are plated on agar medium in order to perform microbiology count and compared it with the inoculated yeast suspension [42]. As for bacteria, the lowest concentration of the drug where a reduction of 99.9% of the initial inoculum is observed is the MFC (Figure 1). 

The lack of standardization has deeply affected the further development in clinical practice of the MFC test. To this point, back when the MFC was beginning to be investigated, Pfaller and colleagues [33,44] reported that the starting inoculum was the first difficulty to overcome in order to evaluate antifungal fungicidal effects in in vitro tests for *Candida* spp. Such criticism was understandable, as the reference document at the time was the M27-A2 procedure, where a concentration range of 0.5 × 10^3^ to 2.5 × 10^3^ CFU/mL was the suggested final inoculum for yeasts. In the first studies of antifungal fungicidal activity, 10 µL from each well at the broth microdilution test were plated [45]. However, according to the final inoculum, the total amount of viable cells was 100 to 500/well, thus a reduction of only >90% could be evaluated. This percentage threshold, as stated by many authors [42,43], could lead to major errors when interpreting the fungicidal activity of antifungal compounds [42]. Based on these assumptions, an early attempt to evaluate fungicidal activity of antifungal molecules conducted by Cantón et al. [43] was to raise the final inoculum to 10^4^ CFU/mL for *Candida* spp. for MFC determination along with an increment in the quantity of yeast suspension plated on a solid medium. In their study, the total amount of broth suspension from clear wells (200 µL) was plated on Sabouraud dextrose agar. The antifungal subject of the study was amphotericin B, and its killing activity was tested against several *Candida* spp. Interestingly, in this experiment, data regarding MIC values for each tested species fell within a close range of values, whereas MFC values expanded into a broader range (76% of isolates presented MFC ≥ 2 × MIC). This was particularly visible for *C. parapsilosis*, which is known to show higher tolerance values for such an antifungal compound [43,46]. This study from Cantón and colleagues [43] was the first attempt to propose a more precise fungicidal test method to determine AMB MFC for several *Candida* spp. In another study, conducted by the same group, under the same testing conditions for MFC, the authors opted to conduct experiments with a higher final inoculum (2.5 × 10^4^ CFU/mL). The range values for MFC were wider for *C. parapsilosis*, *C. glabrata*, and *C. dubliniensis* than those seen for *C. albicans* [47]. 

Other studies worthy of mentioning were conducted by Ernst et al. [44] and Barchiesi et al. [45]. The authors searched for MFC values for micafungin against *Candida* spp. and for caspofungin or AMB against *C. glabrata*. MFC was defined after plating wells above the MIC recovered from the broth microdilution method (according to CLSI M27-A2), where no visible growth was observed. Both groups considered MFC the corresponding antifungal concentration where no colonies were recovered on solid medium after a 48 h incubation [48,49].

An intense field of research where fungicidal tests were applied, such as MFC, evaluated the comparison between different classes or between different antifungal molecules within the same class [50,51]. Like in the case of anidulafungin, micafungin, and caspofungin effects on *C. krusei* [50], the authors evaluated MFC fold dilutions from MIC values. The range of MFC values between the different molecules and MFC to MIC ratios (MFC/MIC) showed that anidulafungin was the most effective echinocandin, as this drug presented the lowest dilution of MFC compared with the other two molecules [50]. Another interesting observation in the different methodological approach taken from this study was related to the final amount of yeast suspension plated on solid medium, which was arbitrarily set at 0.1 mL. As for previous studies, the MFC was determined by looking at the lowest antifungal concentration, but in this case, the authors reported MFC values as those where a reduction of 99.0% of the final inoculum (resulting in less than one colony per agar plate) was observed [50].

From a clinical perspective, little is known about the impact and correlations of MFC values with therapy outcomes of invasive fungal infections. A study from Hirai and colleagues [52] evaluated real-life data obtained from a retrospective survey of *C. parapsilosis* isolated from blood cultures in a tertiary care center from 2000–2010 where both broth microdilution antifungal susceptibility tests and MFC were performed. In their work, the authors investigated the relationship between MFC and MIC of the antifungal compounds tested in routine examination and found that for both amphotericin B and micafungin, but not for fluconazole, the MFC values correlated with the MIC values. Despite these interesting findings, the authors stated that the relationship between MFC and clinical outcomes should be further investigated before drawing firm conclusions [52]. In this study, MFC was defined as the concentration at which no colonies were recovered after plating well suspensions on agar solid medium [52]. 

Despite these clinical evaluations on real-life data isolates, MFC determination is routinely performed when evaluating the antifungal effects of new potential molecules as a screening assessment in the pre-clinical investigations [53,54,55] as well as with new clinically relevant species like *Candida auris*. Evaluation of the fungicidal activity of currently available antifungal drugs in the context of multi-drug-resistant yeast pathogen infections, like *C. auris*, mirrors the usefulness of such tests in supporting clinical therapeutical decisions [51]. Specifically, Duduik et al. [51] reported that echinocandin had no fungicidal effect on 50 strains of *C. auris*, while AMB showed fungicidal activity. Unfortunately, for most *C. auris* isolates, MFC values were found to be ≥32-fold MIC, thus defining a tolerant state of *C. auris* with regard to AMB [51]. 

Different limits have been reported in literature according to MFC determination, consistent with the issues related to MBC determination [42]. First, technical issues, such as the final inoculum, varied from one study to the other, with an arbitrary choice. The final inoculum, suggested by Cantón, of at least 10^4^ CFU/mL, seems to be a valid choice to determine the highest reduction rate (99.9%). In addition, the growth phase of fungal isolates is a major technical issue that should be discussed. Even if it has been previously described with the methods reported, it should still be pointed out that when performing fungicidal activity of antifungal drugs, microbiologists should analyze isolates within the logarithmic phase. This is due to the fact that the stationary phase may include a higher number of dormant and persister cells that would increase the count of living cells after drug exposure [42,56]. A second limit to the standardization of MFC determination is antifungal carryover. Such a phenomenon is associated with the presence of an antifungal compound on solid medium, and it is directly correlated with the plated amount of suspension/well varying across different studies. The carryover of the drug might still exert antifungal activity after plating, inhibiting cell growth and thus lowering MFC values, resulting in an overestimation of the MFC itself. One way to overcome such a problem is to delay the time of streaking after suspension deposition on the plate for at least 20 min in order to allow drug adsorption by the solid media. Again, the time refers to the arbitrary choice of the microbiologist, as no standardization has been appointed so far [42,57]. As previously mentioned, another issue is the arbitrarily chosen amount of suspension plated. If a higher volume of 100 µL is correlated with a more precise colony count and an appropriate evaluation of the required 99.9% reduction from the final inoculum, it has also been associated with an increase of antifungal carryover, whereas a 10 µL volume withdrawn reduces the chances of antifungal carryover but might not be sufficient to calculate the 99.9% reduction needed to assess MFC [42,43].

The third major drawback, aside from the lack of standardization reported above, lies within the fact that like in the case of bactericidal tests, the 99.9% reduction in viable cells has been arbitrarily chosen, and clinical validation of such a correlation with this threshold requires large multicentric studies, which are still few in number. Reports on the correlation between MFC and clinical outcomes have been provided by Nguyen et al. [58], who evaluated 105 patients with candidemia across three different tertiary care hospitals in North America and found an association between higher MFC values at 24 and 48 h and the risk of developing microbiologic failure, defined as positive blood cultures after three days of therapy with AMB. Another multicenter study performed in Brazil by Hartz Alves and colleagues [59] aimed at investigating the correlation between patients’ survival and MFC values of amphotericin B during candidemia episodes. In this case, the authors concluded, as in the previous study, that MFC values might play a more important role in predicting patients’ survival compared to MIC, especially in immunocompromised patients [59]. On the contrary, the above-mentioned study from Hirai and colleagues [52] did not provide analog results. Finally, it is important to discuss a study conducted by Gamaletsou and colleagues [60]. In this case, the authors evaluated breakthrough candidemia episodes in patients affected by hematological malignancies. A major risk factor driving the emergence of breakthrough candidemia was obviously the development of resistance to the antifungal compound currently used to treat the invasive infection. However, the authors also performed MFC determination and found out that an elevated MFC/MIC ratio could be a possible underlying mechanism in explaining breakthrough candidemia, especially in the case of patients receiving intravenous Liposomial AMB and in all cases of *C. parapsilosis* candidemia treated with echinocandin, despite full sensitivity in the in vitro antifungal susceptibility test previously performed [60]. This was an interesting study since it was one of the few clinical studies providing new insights on the correlation between an in vitro fungicidal test and a clinical outcome such as the development of breakthrough candidemia, highlighting the usefulness of such a test beyond common antifungal susceptibility testing. 

The fourth major issue, which is mandatory to discuss, is interpretation of the MFC values in relation to the MIC, which is still debated. Up to this point, it has been defined that isolates with MFC values ≥ 32-fold MICs are to be considered as “tolerant” against the tested agent, even if other authors proposed eight-fold MIC as the threshold to establish tolerance [42]. In addition, another classification that has been addressed by other research groups is to consider an antifungal molecule as fungicidal when the corresponding MFC values resulted to be ≤4-fold MIC, fungistatic when 4 < MFC < 32, and tolerant if MFC ≥ 32-fold MIC [51,61,62]. 

## 3. Time-Kill Curve Analyses

Time-kill curve (TKC) kinetic studies have been extensively studied over time in relation to both antibacterials and antifungals. Since the procedure showed a minor lack of standardization, it is more frequently applied than MFC determination in preclinical and in vitro studies [42]. The concept behind such a method is to evaluate the killing effect by counting the amount of colony reduction in relation to the final inoculum. Such a comparison is done over multiple time points. In this test, microorganisms are cultured in liquid media with different antimicrobial concentrations, i.e., multiples of the MIC. This defines the kinetics of the killing of a single antimicrobial agent over time, and multiple MIC antifungal concentrations are tested simultaneously each in single broth suspension and plated regularly over time (e.g., 0–4–8–12–24 and 48 h depending on the microorganism and the antimicrobial agent). The primary aim of such a method is to find the exact MIC value at which 99.9% or 3-Log unit reduction in the CFU count of the final inoculum after a 24 h incubation is registered [42,63]. In detail, according to the procedure described by Klepser et al. [63,64,65], a time-kill curve analysis usually follows an antifungal susceptibility test that identifies MIC values for the pathogen and the selected molecule. The first step of the procedure is to perform two subsequent subcultures on solid media of the isolate and create a yeast suspension in 9 mL of sterile water in order to obtain the final suspension of 1 × 10^6^−5 × 10^6^ CFU/mL. Next, an aliquot of this suspension is diluted 1:10 in RPMI 1640 liquid medium with MOPS buffer added. Multiples of MIC values of the antifungal drug will be added, incubated for 24–48 h on a rotary wheel, and plated at different time points to assess colony count [63,64,65]. The reproducibility of this protocol was assessed in a multicenter study [65]. A reduction below 99.9% compared to the final inoculum has been associated with a fungistatic effect [42]. A clinically relevant observation that can be drawn from TKC experiments is the lowest MIC value that reaches a fungicidal effect in the shortest amount of time (Figure 2). 

In 2004, Cantón and colleagues [47] provided a very useful mathematical method to evaluate the killing kinetics of antifungal compounds at different concentrations in order to standardize comparison between the same antifungal and different *Candida* species. The formula (log *Nt* = log *N_0_* + *Kt*) is an exponential equation evaluating the amount of viable yeast at certain time points (*Nt*) with the amount of viable yeast in the final inoculum (*N_0_*). *Kt* indicates killing as fungal load in relation to time (CFU/mL/h) [50,61]. Antifungal carryover was reported as a reduction of >25% in the number of colonies between the growth control and the sample added with the antifungal at time zero [63,66].

Despite a standardized protocol, the critical technical issues of time-kill curve analysis are as follows: (i) the logarithmic growth phase of the fungal pathogen should be used in the experiment; (ii) the delivery of the inoculum should be performed as a subsurface delivery of the fungal suspension; (iii) the stability of the antifungal compounds might be reduced within 24 h of the experiment [42].

Time-kill curve analyses have been extensively used in pre-clinical studies: first, to assess the fungicidal effects of newer drugs [48,67,68]; second, to evaluate the potential synergism between drug molecules on resistant isolates [69]; third, to assess the fungicidal effect of the same compound against different yeast species [47,70,71] or to compare the fungicidal effect of two different molecules on the same species [49,64,66]. 

Further studies have investigated the possibility of expanding time-kill curve studies beyond evaluation of the fungicidal/fungistatic effect of the antifungal drug to a deeper insight on antifungal tolerance. Pappalardo et al. [72], on the evaluation of the fungicidal effect of AMB on *Cryptococcus neoformans*, reported that a reduction of 95.9–99.8% in CFU of the final inoculum was associated with the presence of tolerance. 

It would be interesting to evaluate the correlation between time-kill curve analyses and response to therapy. Unfortunately, these studies are mainly related to animal models; human studies consist only of case reports and are still seldomly reported. According to the study conducted by Kardos et al. [73], in vitro antifungal susceptibility and fungicidal test results did not correlate with survival of mice models of *C. krusei* systemic infections. In contrast, in a case of human spondylodiskitis, Pemán et al. [74] found a correlation between clinical failure and MFC and time-kill curve results. In their case, a *Candida krusei* (currently renamed *Pichia kudriavzevii*) spondylodiskitis, presented in an acute myeloid leukemia patient, showed full susceptibility to AMB during routine antifungal susceptibility tests (MIC ≤ 1 mg/L, performed with M27-A2 document, using Sensititre YeastOne and Etest), whilst MFC and time-kill curve analysis showed respectively high fungicidal concentration values (16 mg/L, corresponding to four-fold MIC) and no killing effect (reduction of 95% of viable cells of the final inoculum) after 24 h of incubation at 8 mg/L. Interestingly, the authors correlated poor clinical results of AMB treatment with the in vitro results of fungicidal tests, switching therapy to caspofungin plus voriconazole with an image-documented positive clinical response [74]. Thus far, this is the only case where application of fungicidal determination proved to be more useful than MIC evaluation, imposing reconsideration of the role of such fungicidal tests in clinical laboratory practice for selected cases.

## 4. Serum Fungicidal Concentration

It is widely accepted that therapeutic drug monitoring (TDM), evaluating the serum concentration of antifungal drugs, for example, in the case of voriconazole, provides clinicians with useful information in order to prevent adverse events, and it is also associated with better clinical outcomes in invasive fungal infections [75]. Plus, antimicrobial pharmacodynamics and pharmacokinetics in the case of invasive infections, especially in the context of associated sepsis, might be altered and could be distant from the clinical pictures observed in pre-clinical studies [76]. Thus far, in vitro antifungal and/or antibacterial susceptibility testing have given a general interpretation of the susceptibility profile of a single microorganism from a microbiological perspective [14]. However, clinical resistance and failure of antimicrobial therapy may need other adjunctive factors to be contemplated. Such factors are as important as the in vitro standardized antimicrobial susceptibility test. First and foremost, there is the influence of human body fluids and associated proteins on the bioavailability and pharmacodynamic of the drug [77,78,79]. In this context, as previously cited, serum TDM might be useful to predict drug concentrations in such anatomical sites, yet correlations between serum drug concentrations and clinical outcomes and adverse events have not been fully established for all antifungal compounds. As serum ranges have been validated only for some azoles, serum TDM cannot be applied for other antifungal molecules [80,81,82,83]. Same concepts and difficulties have been reported in the antibacterial field. From such clinical microbiology fields, antifungal research should translate and import important tests that could improve laboratory evaluations of critical cases, such as the serum bactericidal concentration (SBC) [36,38,84]. This technique could be a specific tool from which clinicians might benefit the most, improving antimicrobial therapy and overcoming those black dots where TDM is not readily available or recommended. In addition, it can be useful for patients with selected invasive and difficult-to-treat infections who are experiencing clinical resistance, despite demonstrated in vitro susceptibility of the microorganism to the chosen drug [36,38]. Considering the readily available scientific research on SBC and its clinical application and correlation, it would be useful to apply the same approach to fungal infections.

Serum bactericidal concentration, also known as serum bacterial titers or serum bactericidal activity dates back to 1952 with the study by Fisher and colleagues [84]. Before the introduction of standardized in vitro susceptibility tests and the availability of more effective antibacterial drugs, such a test was the referral method driving anti-infective therapy [36], yet its current application rarely exceeds research purposes, finding its place in peculiar clinical scenarios, such as difficult-to-treat, biofilm-related, multi-drug resistant invasive chronic infections [38]. Briefly, the method reported by Fisher and colleagues [84] is a microdilution titration method like common antifungal susceptibility testing performed with scalar dilution of a patient’s serum obtained after antimicrobial drug administration. The serum of patients is obtained both at peak concentration, within sixty minutes after the end of infusion at maximum, and trough concentration, evaluated immediately before delivery of the subsequent dosage of the drug. Results from a microdilution titration method define the serum drug concentration inhibiting microbial growth, thus defining the serum bacteriostatic concentration of the currently administered drug in regard to the clinical isolate, causing the underlying invasive infection. The aim of the SBC test is to evaluate the lowest serum dilution at which ≥99.9% or 3-Log of the initial bacterial inoculum (5 × 10^5^ CFU/mL) is killed by subculturing on solid agar media serum and yeast suspensions from wells where no visible growth was detected in the serum bacteriostatic concentration test (Figure 3) [36,84]. 

Based on these assumptions, it appears clear that one of the major drawbacks of such a method is related to keeping the isolated microorganism causing the infection and obtaining the proper amount of a patient’s serum at the right time, which may significantly vary based upon the pharmacokinetic properties of different antimicrobial drugs and also processing it in the shortest time possible. This is due to the reduced stability of the compound in patient’s serum after sampling [85]. Therefore, assessing the SBC might be unfeasible to perform in routine workflow in clinical microbiology laboratories and should be prioritized only in selected patients and tertiary-level healthcare centers [36,86]. To this point, a high degree of laboratory expertise is also required when it comes to interpretation of results. In the case of antibacterial drugs, dilutions at which positive clinical outcomes have been registered vary based upon the selected antimicrobial agent and the corresponding clinical condition (e.g., SBC titers associated with positive clinical outcome are not the same for endocarditis or osteomyelitis caused by the same microorganism) [36]. 

A similar approach to the SBC technique has been reported by Stein and colleagues [87] for yeast. They evaluated fungicidal effects of peak and trough anidulafungin serum concentrations obtained from patients that were treated for suspected or proven candidemia. However, the aim of the study was to evaluate the effect of human serum protein bound with anidulafungin on the killing effect of the drug against different *Candida* spp. that were not the causative agents of the patients’ infectious disease. Still, an important message could be obtained from this study, as peak concentrations of anidulafungin exhibited a killing effect only on yeast isolates with low MIC concentration (<0.1 mg/L) on the in vitro antifungal susceptibility test [87]. However, serum fungicidal concentration (SFC) determination still remains an unexplored field with no referral guidelines and no solid correlation established between SFC titers and clinical outcomes. SFC would grant beneficial adjunctive information especially in the case of chronic biofilm-related invasive fungal infections, especially in the context of increasing azole-resistance and multi-drug resistance among yeast isolates.

A list of advantages and disadvantages for all phenotypic fungicidal tests has been made in Table 1. 

## 5. Microbiological and Pharmacological Factors Altering Interpretation of Fungicidal Tests in Clinical Practice: Persistence, Tolerance, Paradoxical Growth, and Post Anti-Fungal Effect

In vitro fungicidal tests have been extensively studied through time, although they have not found their place yet in clinical microbiological routines. Many reasons account for such conditions apart from technical issues, and a high degree of expertise is required for laboratory staff. 

The first reason relies on the biology and microbiological characteristics of the pathogen, in particular to the presence of persister cells within the colonies picked for antifungal susceptibility testing. The second reason is due to the lack of standardization at the in vitro test protocols [42]. As the technical issues have been previously discussed within each technique, the presence of persister cells can be generalized to each method described. Indeed, both bacterial and fungal microorganisms included in a single colony may exhibit peculiar phenotypes, such as the persister cells commonly found in microbial biofilms [88,89]. These cells can be distinguished by other microorganisms present in the same population according to their reduced metabolic activity [90,91]. Such physiological conditions reduce the effect of drugs on the microorganism itself without the development of actual resistance. This phenomenon is represented by the growth of microbial colonies in the trailing growth during broth microdilution tests or on solid medium for both disk diffusion and E-test strips. [42,91,92]. Persistence has also been observed in *A. fumigatus* in relation to its fungicidal drug voriconazole [93]. Based on the findings reported in their study, Scott et al. [93] suggest that persistence may play a potential role in treatment failure for voriconazole in *A. fumigatus* infections. 

As for bacteria, fungi have also shown tolerance to antifungal agents [35]. Tolerance is defined as the slow growth of susceptible fungal strains under concentrations of drugs above the MIC, and the authors have highlighted that tolerance to antifungal drugs varies upon both the tested agent and strain, with different grades from low to high tolerance levels [25,35]. Tolerant microorganisms have not been associated with an underlying resistance mechanism, as re-testing of such pathogens for the same antifungal molecules have not resulted in increased MICs and/or display of resistant phenotypes at routine antifungal susceptibility tests. Still, in vitro growth above MIC values is seen with prolonged incubation of the yeast (≥48 h) and is depicted by the “trailing growth” in broth microdilution tests or fraction of growth within the MIC inhibition zone diameter in disk diffusion assays, which can be calculated using the *diskImageR* software (Version 4.3.2) (latest version downloadable at https://cran.r-project.org/) [25,94]. Even E-tests can provide a visualization of antifungal tolerance confronting the degree of growth between colonies within and outside the inhibition zone of the antifungal [25]. It has been reported for *C. albicans* [25] and *C. parapsilosis* exposed to fluconazole and echinocandin, respectively [19,95]. Since it has not yet been directly associated with in vivo development of resistance and/or therapeutic failure, some authors even question its clinical implication [25]. The previously mentioned persister cells have been appointed as one of the possible explanatory factors behind the development of antifungal tolerance. Other than that, increased chitin synthesis, activated cellular stress response-molecular pathways, and changes in the sphingolipid composition of the fungal membrane have all been pointed to as contributing factors to the development of tolerance in fungi [12,25,35]. Still, despite the current debate on their clinical relevance, persister cells and tolerance could result in altering the interpretation of in vitro fungicidal tests [42].

Another major issue that is microorganism-related and might alter the results interpretation and evaluation of in vitro fungicidal tests is the paradoxical growth or “eagle effect” [96,97,98]. Briefly, such an in vitro phenomenon has been observed in both *Candida* spp. and *Aspergillus* spp. when evaluating susceptibility to echinocandin, and it depicts the ability of the pathogens to grow in broth microdilution methods under elevated drug concentrations, higher than MIC values, while maintaining full susceptibility at lower concentrations of the same antifungal [97,98,99]. Such paradoxical growth has been shown to be species-specific with *C. tropicalis* and *C. albicans* as the most frequent species presenting the eagle effect [100]. This paradoxical growth is also driven by echinocandin drugs, with caspofungin as the molecule showing the strongest association [99]. In addition, paradoxical growth is not associated nor influenced by prior exposure to echinocandin, as it has been observed even in drug-naïve isolates [101]. Moreover, it has not been correlated to treatment failure [102]. Such conditions could be explained, as the drug concentration levels obtained in vivo might not be as high as those that trigger this in vitro phenomenon [101,103]. However, to this point, a study from Binder et al. [104] is worthy of mentioning. Authors found paradoxical growth of *C. albicans* in an in vitro test for caspofungin to be associated with variable treatment outcomes in a *Galleria* model of infection. 

Other contributing factors that might influence the development of the eagle effect are culturing conditions. Indeed, it has been reported that antifungal susceptibility tests conducted with RPMI showed a reduced tendency to induce such paradoxical growth in *C. albicans*; plus, when this culture media was added with serum at a concentration of 50%, the eagle effect was abolished [97,99]. Growing conditions of the yeast pathogen, sessile or planktonic, seem to affect the development of the eagle effect. In particular, when isolates are grown under biofilm conditions, they show a higher tendency toward paradoxical growth. This is due to the fact that sessile cells within biofilm present a different metabolic profile when compared to planktonic counterparts [99,105]. This change in the metabolic activity of target cells, as seen in persister cells, might be the explanation of such paradoxical growth [106]. Several authors have investigated the molecular reason for such a phenomenon within the yeast genome. The results did not link the eagle effect to a newly acquired genetically driven resistance mechanism, such as *FKS* 1–2 point mutations, nor the overexpression of target genes or drug inactivation [101]. The paradoxical growth is therefore associated with and partially explained by adaptation to antifungal drugs. One way to manage this is to increase cell-wall chitin content, as hypothesized and demonstrated by Stevens et al. [107], Rueda and colleagues [101], and Bizerra et al. [108]. Finally, exposure of isolate to caspofungin prior to performing the antifungal susceptibility test on the selected echinocandin [109] or adding Nikkomycin Z to the 50% human serum while testing for echinocandin susceptibility [110] are two more ways to avoid paradoxical growth. 

Another issue that interferes with fungicidal test interpretation reported for several antifungal compounds is the prolonged inhibition of fungal growth lasting even hours after drug exposure [111], as previously observed for bacterial isolates [112]. First described by Ernst and colleagues [111], this phenomenon, called the post-antifungal effect (PAFE) was initially assessed, exposing fungal isolates to antifungals for a short period of time. In their study on *C. albicans*, Ernst et al. [111] exposed the microorganism at different concentrations of three major antifungal classes, such as azoles, echinocandins, and polyenes, for a time span of 15 to 60 min. Then, three consecutive antifungal removal procedures by centrifugation and washing were performed, obtaining a final pellet that was resuspended in liquid medium. Such a fungal suspension was then incubated; next, aliquots were taken and plated on solid medium at serial time intervals in order to evaluate colony count. This standardized procedure was adopted in other experiments where changes were applied only to incubation time after disposal of the fungal suspension on solid medium according to the species-specific time of growth [113,114]. Results from multiple studies conducted on PAFE showed that this phenomenon was largely influenced by the species, the class of molecule, as AMB and echinocandin showed prolonged inhibition of growth after removal, and drug concentration, as higher PAFE was registered with higher antifungal concentrations [114,115]. 

From a clinical and pharmacological standpoint, PAFE influences the timing of antifungal drug administration, with drugs with prolonged PAFE requiring a lower dosing per day [111,115]. On the other hand, PAFE-induced prolonged inhibition of growth might alter the results of fungicidal in vitro tests, as microorganisms whose growth was only PAFE-inhibited were not actually killed. Therefore, such a phenomenon might lead to overestimation of the killing rate. To avoid such a problem, the authors recommend prolonging the incubation time after plating and avoiding antifungal carryover on the solid medium. To do so, the operators must wait until the disposed aliquot on the plate is dried to perform streaking of the dispensed fungal cells [42,111]. 

## 6. Other Non-Killing In Vitro Assays: Detection of Antifungal Tolerance in Yeast

Tolerance is defined as the capability of slowly growing during in vitro susceptibility tests under drug concentrations above the MIC, the so-called “supra-MIC growth,” of a sub-proportion of microorganisms, without any known genetic determinants of antimicrobial resistance [25,35,94,116]. Such phenotypic-related slow growth has been linked to the ability of the pathogen to withstand the drug-induced cellular damage instead of development of resistance mechanisms [117,118]. As stated previously, the sub-population of microbial cells developing the supra-MIC growth can be related to the presence in the final inoculum of persister cells, also known as “dormant cells,” showing reduced metabolism and thus altering the effect of the antifungal drug [119]. Tolerance appears to be more pronounced when the tested antifungal exerts only a fungistatic effect, with azoles as the most frequently reported compounds to be associated with supra-MIC growth, also defined as “trailing growth” in broth microdilution tests [118]. Moreover, antifungal effects of different molecules are species-specific, as in the case of *Candida parapsilosis* with the echinocandin class. These compounds, mainly fungicidal, were proven to have only a fungistatic effect on *C. parapsilosis* due to a naturally occurring polymorphism in the target enzyme (P660A in the hot spot region 1 of the subunit Fks1p) [120]. To date, the clinical impact of antifungal tolerance on the development of acquired resistance is still a matter of debate. Since it has not yet been linked to in vivo demonstration of the actual development of resistance, several authors and studies have reported an association at in vitro experiments, as higher degrees of tolerance are predisposing factors for the in vitro acquisition of resistance [19,25,95,121]. This has led to speculation that evaluation of antifungal tolerance should be assessed more frequently, especially in the case of outbreaks of invasive fungal infections with acquired resistance, as isolates with higher levels of tolerance in in vitro tests have been associated with persistent candidemia [35]. Recent outbreaks of azole-resistant *C. parapsilosis* are the most representative examples. In such clinical scenarios, therapeutic options are hindered, and the continuous use of a single antifungal agent like an echinocandin might exert the selective drug pressure needed to select resistant strains [19,35,95]. Such clinical conditions could be the best testing ground for the hypothesized correlation between different grades of antifungal tolerance and the actual development of in vivo resistance. Still, another unmet need in current clinical practice is to establish whether there is a correlation between different levels of antifungal tolerance and clinical outcomes in patients undergoing therapy [25,118]. 

To determine antifungal tolerance, two different methods have been proposed, i.e., disk diffusion and broth microdilution [25]. Usually, the incubation time is longer than that required for MIC evaluation. Disk diffusion-based quantification of drug tolerance of fungal isolates is based on the application of the previously mentioned standardized procedure *diskImageR* software [94] (Figure 4). 

Such program is a free downloadable software extension of the R program and simply calculates pixel intensity of the uploaded photos of disk-diffusion tests according to several radial lines (72 in total, one every 5° for the total 360° of the image) drawn from the disk to the periphery of the inhibition zone. Pixel intensity reflects and estimates cell density; therefore, the software works by comparing mean pixel intensity within the zone of inhibition along each point of the drawn line with the mean pixel intensity outside the zone inhibition, where fungal pathogens have grown without being affected by the antifungal. By such calculations, the program then estimates three distance measures from the edge of the disk, defined as the radius of inhibition (RAD), where fungal growth is reduced by 20, 50, and 80% (RAD_20_, RAD_50_, and RAD_80_, respectively). In their initial study, Gerstein and colleagues [25,94] used RAD values to infer resistance and interpret disk diffusion susceptibility tests according to RAD values. Specifically, RAD_20_ is the point where fungal growth is reduced by 20%, and it defines the radius and therefore the distance from the disk where a resistant fungal population starts to appear. It is clear that if RAD_20_ falls within the pre-defined inhibition zone diameter, the tested strain will have to be reported as resistant. Secondly, by plotting pixel intensity in relation to the distance from the disk edge and calculating the area under the curve by every RAD point, the *diskImageR* software calculates the fraction of growth (FoG) within the inhibition zone of the disk diffusion, allowing for the direct estimation of antifungal drug tolerance. The authors have assessed and established a FoG of 50% as a tolerance cut-off point. The estimation of tolerance using the FoG can be performed by assessing the value of FoG for each RAD included in the zone of inhibition. Such estimation, as previously stated, is based on the direct calculation of the area under the curve of the graph fitting the average pixel density and corresponding RAD values [25,94].

The broth microdilution method isanother way to assess tolerance. In this case, tolerance is associated with trailing growth, which is the microbiological evidence of supra-MIC growth (SMG) visible after 48–72 h of incubation. Despite being visible to the naked eye, such a phenomenon can and should be standardized with routine examination assessed via optical density measurements. These measurements aim to define the ratio between the average optical density of each well where growth of tolerant isolates has been observed and the optical density value of wells where no growth was observed [25]. 

Lastly, the presence of tolerance to antifungals can also be inferred from E-test strips after prolonged incubation. Despite being easy to assess, as the growth within the inhibition zone is feasible to evaluate without the help of any instrument, such a method does not allow for a precise quantification of the degree of tolerance. Even if authors assess that different levels of visible cell density are found within the zone of inhibition, the E-test strip has not yet been provided with a direct quantification tool for detecting and assessing degrees of tolerance. As a consequence, microbiologists could only report the presence or absence of tolerance to the selected antifungal, and tolerance can be described as a categorical variable [25].

## 7. Miscellaneous

With this review, we wanted to focus specifically on in vitro non-molecular phenotypic fungicidal test assays sharing similar methodological approaches to those currently applied in clinical practice in the evaluation of MIC values. Correlation between fungicidal values of antifungal molecules and clinical outcomes in the case of invasive yeast infections harboring antifungal resistance has not yet been determined. However, it is mandatory to mention other diagnostic strategies that are potentially useful in the context of emerging antifungal resistance [122]. First, the most promising molecular method is matrix-assisted laser absorption deionization time of flight (MALDI-TOF). This proteomic method has been studied with machine learning in order to deliver rapid assessment of fluconazole resistance in *C. albicans* [123]. In addition,, other authors have investigated the effects of different concentrations of antifungal drugs on the mass spectrometry profiles of *Candida* spp. isolates compared to untreated strains. Based on the lowest concentration of antifungal drugs altering the MALDI-TOF spectra, authors were able to draw information regarding the correlated susceptibility profile of the *Candida* spp. isolate [124,125,126].

Other non-proteomic-based tests that evaluate susceptibility profiles of yeast pathogens are the flow cytofluorometric method, ATP bioluminescence assay, and thin-layer chromatography (TLC)–bioautography [33,122]. Despite the latter not being a labor-intensive procedure, it does not provide information on fungicidal activity of antifungal molecules. ATP bioluminescence assay estimates fungal load and vitality by assessing the production of adenosine triphosphate (ATP) by microorganisms within cultures [122]. Lastly, the flow cytofluorometric method detects direct cellular damage and viability caused by the selected antifungal compounds. For this technique, faster and reproducible results are easily obtained, inferring even fungicidal effects of antifungal drugs; however, a limitation to its implementation in clinical practice is the high technical expertise required to use the cytofluorometer itself [33].

## 8. Discussion

Determination of drug fungicidal concentration via MFC, time-kill curve analyses, or SFC has not found its place yet in routine clinical practice, despite being utilized over a relatively long period of time for research and/or preclinical purposes. In vitro fungicidal tests still lack in vivo correlation and some aspects of standardization. As these aspects might resemble a major drawback in the further development and utilization of such methods in clinical practice, they might also ironically be the reasons that discourage authors from exploring these research fields at the same time. Moreover, in vivo correlations studies are not easy to assess, as they require trained personnel due to the technical difficulties within such analyses. Apart from that, it is also important to mention that killing effect can only be evaluated for fungicidal drugs. However, in clinical practice, most antifungal drugs used are fungistatic, and, therefore, a restricted group of drugs and pathogens can be the subject of prospective fungicidal test studies. Plus, from a cost-effective point of view, considering the amount of work and resources implied in performing such tests, it will not be feasible to apply them for all kinds of infections, but their clinical role should be narrowed to only difficult-to-treat invasive fungal infections, like endocarditis, meningitis, osteomyelitis, joint and prosthesis infections, and deep-seated ocular and ear infections. To this point, considering (i) the raising concern about acquired antifungal resistance and the continuous reports of clonal outbreaks of azole- and echinocandin-resistant *Candida* spp. invasive infections and (ii) the narrowing of the therapeutic options, it might be useful to also add in vitro fungicidal tests to routine antifungal susceptibility tests. Above all, SFC could give useful information to the clinician even beyond therapeutic drug monitoring, and its correlation with clinical and/or microbiological failure might be worthy of investigation. However, based on the assumptions previously reported, it might not be feasible to put research studies and protocols in practice. They need prospective multicenter studies requiring multiple resources, becoming therefore expensive and time consuming and also implying tight collaboration between laboratory and clinical personnel. 

Despite studies on animal models being used as in vivo correlates, evaluation of the potential association between fungicidal tests and clinical and/or microbiological failure is an unmet need in current microbiology. Apart from the clinical correlation from the study of Nguyen et al. [58] that pointed out that in candidemia patients, MFC data accounted for microbiological failure as its strongest predictor, few reports are available on the efficacy of fungicidal tests in predicting clinical response in invasive yeast infection. To date, fungicidal tests have greatly suffered from a lack of standardization as well as reproducible results in the same species. Nonetheless, they could be depicted as possible adjunctive in vitro tests for all the fungi with no EUCAST or CLSI breakpoints and for those with currently intermediate susceptibility as well as in peculiar clinical settings, such as severe invasive infections, where prolonged therapy might be required, thus leading to the risk of dissemination and treatment failure.

In addition, considering the rising concern about the increase of antifungal resistance from different *Candida* species among clinical isolates all over the world, efforts in research should aim at predicting the development of antifungal resistance, thus preventing its spread. In this context, the role of antifungal tolerance has not been extensively explored. Defining its contribution in the evolutionary pathway towards the development of resistance as well as its potential clinical implication as a standalone risk factor are musts for future research. From our perspective, it would seem reasonable to propose that in the context of outbreaks of antifungal resistant strains, investigation and quantification of tolerance with a standardized approach should be integrated in routine laboratory practice, at least against the recommended treatment (e.g., testing tolerance against echinocandins on bloodstream infection isolates in the case of outbreaks of azole-resistant *C. parapsilosis*) or in any case where tolerance is detected in in vitro susceptibility tests.

## 9. Conclusions

In vitro killing studies for antifungals lack standardization and might be influenced by many factors. Antifungal tolerance is an emerging field of research, and this microbiological phenomenon along with its interpretation and clinical correlation still need to be contextualized in clinical microbiology practice in order to predict and evaluate the emergence of antifungal resistance. Furthermore, we believe that MFC, time-killing curves, and SFC should be adjunctive routine examinations for microbiological laboratories in the case of drug-resistant *Candida* spp. causing systemic infections in order to prevent the development of clinical and/or microbiological failure. 

## 10. Future Directions

The contribution of fungicidal tests in predicting clinical outcomes, especially the implementation of serum fungicidal concentration, is worthy of future investigation within multicenter prospective studies. At the same time, considering the increase in antifungal tolerance, its contribution to the development of in vivo resistance, and its correlation with clinical outcomes is necessary and should be evaluated with large prospective studies. 

## Figures and Tables

**Figure 1 jof-09-01188-f001:**
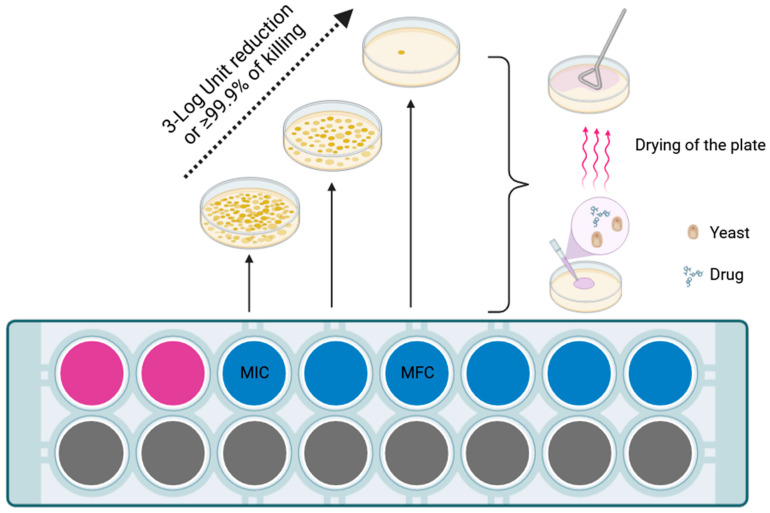
Minimum fungicidal concentration assessment method. MIC = minimum inhibitory concentration; MFC = minimum fungicidal concentration. After determination of the MIC value, yeast and antifungal drug suspensions contained in each clear well were displaced on solid agar medium. To avoid antifungal carryover, streaking is performed once the suspension is soaked and the plate is dried. Plates are incubated at 35 °C for 24–48 h in order to count yeast colonies. The first concentration at which 3-Log unit reduction or 99.9% of killing of the final inoculum is observed corresponds to the MFC. Procedure according to Cantón et al. [43]. Image created with BioRender.com.

**Figure 2 jof-09-01188-f002:**
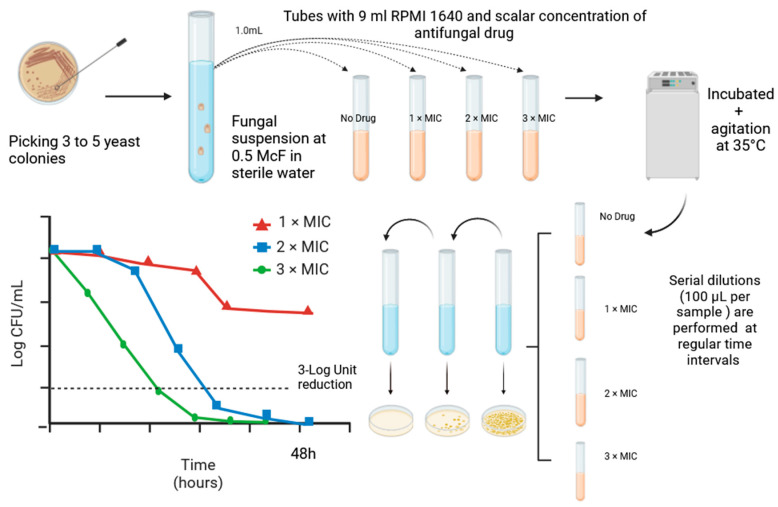
Time-kill curve assessment method. Time-kill curve analysis according to Klepser et al. [63]. Fungicidal concentration is calculated as the lowest antifungal drug concentration exerting a killing effect (3-Log unit reduction in CFU count or 99.9% of killing of the initial inoculum) at the end of the experiment (24–48 h) as highlighted in the line graph on the left.

**Figure 3 jof-09-01188-f003:**
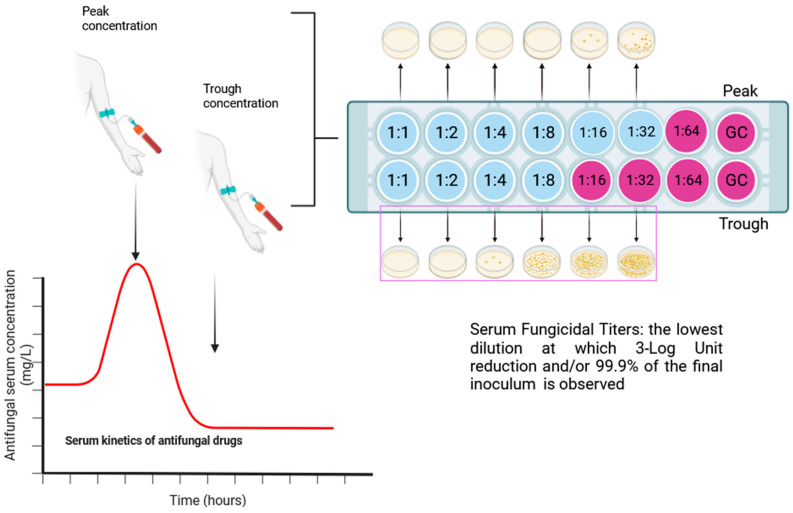
Serum fungicidal concentration assessment method. Schematic representation of the serum fungicidal concentration method. On the right serial dilutions of patient’s serum added with fungal suspension with a final inoculum of 5 × 10^5^ CFU/mL. This method refers to the SBC determination methods proposed by Fisher and described by Wolfson and colleagues [84]. GC = growth control. Red line represents the serum concentration kinetic of the selected antifungal.

**Figure 4 jof-09-01188-f004:**
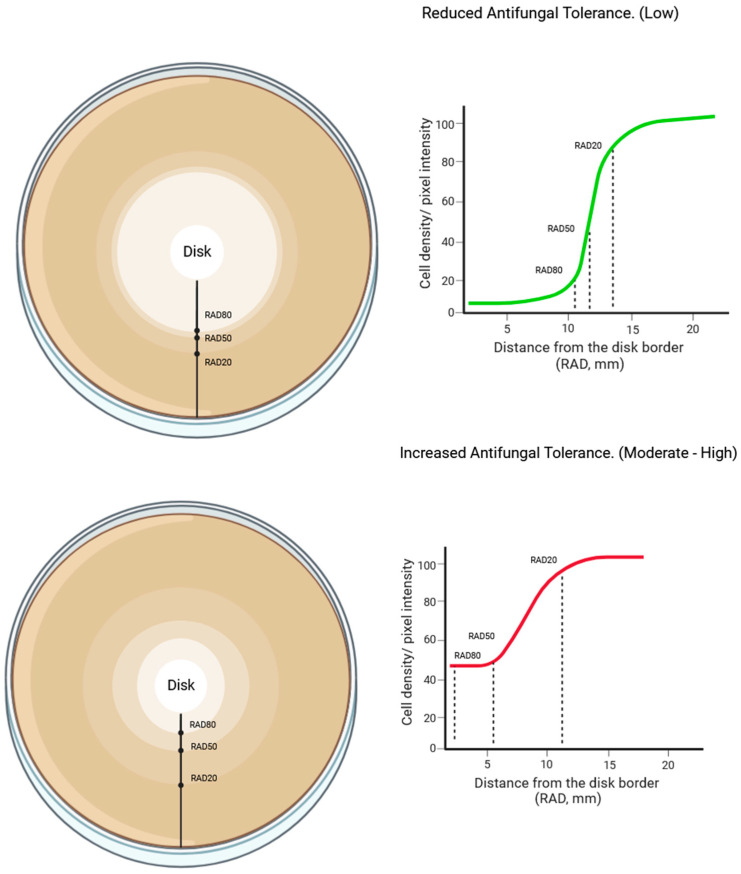
Evaluation of antifungal tolerance during the disk diffusion test using the *diskImageR* software (https://cran.r-project.org/). Cell density is highlighted by different shades of color; a more intense color defines a higher cell density and is associate with a higher pixel intensity. RAD = radius of inhibition. RAD20, RAD50, and RAD80 represent the respective distances from the border of the disk where reductions of growth of 20, 50, and 80% are observed, respectively. The area under the curve estimates the fraction of growth (FoG).

**Table 1 jof-09-01188-t001:** List of advantages and disadvantages for all phenotypic fungicidal tests.

Procedures	Advantages	Disadvantages
Minimum Fungicidal Concentration	Results interpretation; Reduced costs	Standardization of the procedure Lack of clinical correlation
Time-Kill Curve analysis	Standardization of the procedure; Results interpretation	Technical expertise required, Time-consuming Lack of clinical correlation
Serum Fungicidal Concentration	Standardization of the procedure; Reduced cost Results with direct clinical correlations	Technical expertise required; Results interpretation Requires extra blood samples to be taken from patients Analyses to be performed in the shortest amount of time possible due to the reduced stability of the antimicrobial drugs after sampling.

## Data Availability

No new data were generated to perform this review article, all relevant information are reported in the reference section.

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
