# Peer review of "In Vitro Susceptibility Tests in the Context of Antifungal Resistance: Beyond Minimum Inhibitory Concentration in Candida spp."

_jof, 2023, doi:10.3390/jof9121188_

Round 1
Reviewer 1 Report
Comments and Suggestions for Authors
Dear authors,
I read your review concerning in-vitro susceptibility tests regarding yeast in the context of Antifungal Resistance. The review is well written, clear and of unblemished clinical application. Furthermore, several current hot spots are eviscerated, including difficult-to-treat infections, unmet needs, antimycotic stewardship and the underlying fundamental (which I believe should be made explicit as a formal point) inherent to translational research. Despite that, some points should be addressed to improve the manuscript.
1. Interestingly, you didn’t report our previous work about the use of MALDI-TOF MS for the detention of Antifungal Resistance. According to your review, you evaluated the in vitro tests but a focus on this proteomic method should be reported (few lines in introduction). Moreover, it is not clear what criteria did you follow to include tests. Flow cytofluorometric method, ATP bioluminescence assay and Thin-layer chromatography (TLC)–bioautography are missing. Why those methods have been excluded?
2. mL, please
3. reference just after Author et al. [], not at the end of the sentence.
4. Standardize in vitro or in-vitro, please.
5. In the introduction, few lines about One Health and fungal infections should be included. I suggest reading:
- Nnadi, N.E.; Carter, D.A. Climate change and the emergence of fungal pathogens. PLoS Pathog. 2021, 17, e1009503.
- Coates, S.J.; Norton, S.A. The effects of climate change on infectious diseases with cutaneous manifestations. Int. J. Women’s Dermatol. 2021, 7, 8–16.
6. Lines 657-659, but also in all the fungi with no EUCAST or CLSI breakpoints and in those with currently intermediate susceptibility. Moreover, I suggest reading:
- Binder U, Aigner M, Risslegger B, Hörtnagl C, Lass-Flörl C, Lackner M. Minimal Inhibitory Concentration (MIC)-Phenomena in Candida albicans and Their Impact on the Diagnosis of Antifungal Resistance. J Fungi (Basel). 2019 Sep 4;5(3):83. doi: 10.3390/jof5030083. PMID: 31487830; PMCID: PMC6787722.
7. Lines 248-250, I suggest reading and citing:
- Codda G, Willison E, Magnasco L, Morici P, Giacobbe DR, Mencacci A, Marini D, Mikulska M, Bassetti M, Marchese A, Di Pilato V. In vivo evolution to echinocandin resistance and increasing clonal heterogeneity in Candida auris during a difficult-to-control hospital outbreak, Italy, 2019 to 2022. Euro Surveill. 2023 Apr;28(14):2300161. doi: 10.2807/1560-7917.ES.2023.28.14.2300161. PMID: 37022211; PMCID: PMC10283462
8. A table reporting methods, vantage and disadvantages could represent a good way to summarize the review results.
Comments on the Quality of English LanguageMinor editing of the English language is required.
Author Response
Reviewer #1
Dear authors,
I read your review concerning in-vitro susceptibility tests regarding yeast in the context of Antifungal Resistance. The review is well written, clear and of unblemished clinical application. Furthermore, several current hot spots are eviscerated, including difficult-to-treat infections, unmet needs, antimycotic stewardship and the underlying fundamental (which I believe should be made explicit as a formal point) inherent to translational research. Despite that, some points should be addressed to improve the manuscript.
- Interestingly, you didn’t report our previous work about the use of MALDI-TOF MS for the detention of Antifungal Resistance. According to your review, you evaluated the in vitro tests but a focus on this proteomic method should be reported (few lines in introduction). Moreover, it is not clear what criteria did you follow to include tests. Flow cytofluorometric method, ATP bioluminescence assay and Thin-layer chromatography (TLC)–bioautography are missing. Why those methods have been excluded?
We thank Reviewer #1 for his/her useful and knowledgeable work. According to the aforementioned suggestions one statement and a new section have been added to the manuscript. The new section goes under the name “Miscellaneous”:
Line 124. Aim of this study is to revise current in-vitro non-molecular phenotypic fungicidal test assays beyond quantification of MIC.
Line 650: 7. Miscellanous
With this review we wanted to focus specifically on in-vitro phenotypic fungicidal test assays sharing similar methodological approaches to those currently applied in clinical practice in the evaluation of MIC values. Correlation between fungicidal values of antifungal molecules have not been yet correlated with clinical outcomes in the case of invasive yeast infections harboring antifungal resistance. However, it is mandatory to mention other diagnostic strategies potentially useful in the context of emerging antifungal resistance [122]. First, the most promising molecular method is Matrix Assisted Laser Absorption Deionization Time of Flight (MALDI-TOF). Such proteomic method has been studied with machine-learning in order to deliver rapid assessment of fluconazole resistance in C. albicans [123]. Still, other authors instigated the effects of different concentrations of antifungal drugs on the mass spectrometry profiles of Candida spp. isolates compared to untreated strains. It was based on the lowest concentration of antifungal drug altering the MALDI-TOF spectra that authors were able to draw information regarding the correlated susceptibility profile of the Candida spp. isolate [124-126].
Other non preoteomic-based tests that evaluate susceptibility profiles of yeast pathogens are Flow cytofluorometric method, ATP bioluminescence assay and Thin-layer chromatography (TLC)–bioautography [33,122]. Despite the latter is not a labour-intensive procedure, it does not provide information on fungicidal activity of antifungal molecules. ATP bioluminescence assay estimates fungal load and vitality by assessing the production of adenosine triphosphate (ATP) by microorganisms within cultures [122]. Lastly, Flow Cytofluorometric detects direct cellular damage and viability caused by the selected antifungal compounds. For this technique faster and reproducible results are easily obtained inferring even fungicidal effects of antifungal drugs, however a limitation to its implementation in clinical practice is the high technical expertise required to use the cytofluorometer itself [33].
The following references have been added:
- Balouiri, M.; Sadiki, M.; Ibnsouda, S.K. Methods for in Vitro Evaluating Antimicrobial Activity: A Review. J Pharm Anal 2016, 6, 71–79, doi:10.1016/j.jpha.2015.11.005.
- Delavy, M.; Cerutti, L.; Croxatto, A.; Prod’hom, G.; Sanglard, D.; Greub, G.; Coste, A.T. Machine Learning Approach for Candida Albicans Fluconazole Resistance Detection Using Matrix-Assisted Laser Desorption/Ionization Time-of-Flight Mass Spectrometry. Frontiers in Microbiology 2020, 10.
- Marinach, C.; Alanio, A.; Palous, M.; Kwasek, S.; Fekkar, A.; Brossas, J.-Y.; Brun, S.; Snounou, G.; Hennequin, C.; Sanglard, D.; et al. MALDI-TOF MS-Based Drug Susceptibility Testing of Pathogens: The Example of Candida Albicans and Fluconazole. Proteomics 2009, 9, 4627–4631, doi:10.1002/pmic.200900152.
- Sanguinetti, M.; Posteraro, B. Mass Spectrometry Applications in Microbiology beyond Microbe Identification: Progress and Potential. Expert Rev Proteomics 2016, 13, 965–977, doi:10.1080/14789450.2016.1231578.
- De Carolis, E.; Vella, A.; Florio, A.R.; Posteraro, P.; Perlin, D.S.; Sanguinetti, M.; Posteraro, B. Use of Matrix-Assisted Laser Desorption Ionization-Time of Flight Mass Spectrometry for Caspofungin Susceptibility Testing of Candida and Aspergillus Species. J Clin Microbiol 2012, 50, 2479–2483, doi:10.1128/JCM.00224-12
- mL, please
Line 138 (CFU/mL)
- reference just after Author et al. [], not at the end of the sentence.
All references have been corrected accordingly.
- Standardize in vitro or in-vitro, please.
In-vitro was reported all over the text accordingly.
- In the introduction, few lines about One Health and fungal infections should be included. I suggest reading:
- Nnadi, N.E.; Carter, D.A. Climate change and the emergence of fungal pathogens. PLoS Pathog. 2021, 17, e1009503.
- Coates, S.J.; Norton, S.A. The effects of climate change on infectious diseases with cutaneous manifestations. Int. J. Women’s Dermatol. 2021, 7, 8–16.
Line 63. Still, more threats to come are related to the consequence of climate change on human-fungal pathogenesis and interaction. As elucidated in the One health approach by the WHO and meticulously described in the studies by Nnadi et al. [31] and Coates and colleagues [32], challenges from newer fungal species adapting to human biology, as well as yet-undiscovered newly associated resistance mechanisms will arise in the sooner future. Among all, for Candida spp. isolates it has been suggested that human adaptation and infections caused by C. auris, (which represents the paradigm of Multi-Drug Resistant Candida spp.), could be indeed inferred to climate change. Based on these assumptions not only the population of acquiring an invasive fungal infection under climate change pressure is increasing but also the number of species along with their new antifungal-resistance traits able to adapt to human biology are expected to increase [32].
The following references have been added:
- Nnadi, N.E.; Carter, D.A. Climate Change and the Emergence of Fungal Pathogens. PLOS Pathogens 2021, 17, e1009503, doi:10.1371/journal.ppat.1009503.
- Coates, S.J.; Norton, S.A. The Effects of Climate Change on Infectious Diseases with Cutaneous Manifestations. International Journal of Women’s Dermatology 2021, 7, 8–16, doi:10.1016/j.ijwd.2020.07.005.
- Lines 657-659, but also in all the fungi with no EUCAST or CLSI breakpoints and in those with currently intermediate susceptibility. Moreover, I suggest reading:
- Binder U, Aigner M, Risslegger B, Hörtnagl C, Lass-Flörl C, Lackner M. Minimal Inhibitory Concentration (MIC)-Phenomena in Candida albicans and Their Impact on the Diagnosis of Antifungal Resistance. J Fungi (Basel). 2019 Sep 4;5(3):83. doi: 10.3390/jof5030083. PMID: 31487830; PMCID: PMC6787722.
According to reviewer’s suggestions we have added the following statement in the discussion section:
Line 714. for all the fungi with no EUCAST or CLSI breakpoints and in those with currently intermediate susceptibility as well as
A specific mention to the suggested reference has been added to section 5. Microbiological and pharmacological factors altering interpretation of fungicidal tests in clinical practice: Persistence, Tolerance, Paradoxical Growth and Post Anti-Fungal Effect
Line 505-507. However, to this point a study from Binder et al. [103] is worthy of mentioning. Authors found paradoxical growth of C. albicans at in-vitro test for caspofungin to be associated with variable treatment outcomes in a Galleria model of infection.
- Lines 248-250, I suggest reading and citing:
- Codda G, Willison E, Magnasco L, Morici P, Giacobbe DR, Mencacci A, Marini D, Mikulska M, Bassetti M, Marchese A, Di Pilato V. In vivo evolution to echinocandin resistance and increasing clonal heterogeneity in Candida auris during a difficult-to-control hospital outbreak, Italy, 2019 to 2022. Euro Surveill. 2023 Apr;28(14):2300161. doi: 10.2807/1560-7917.ES.2023.28.14.2300161. PMID: 37022211; PMCID: PMC10283462
We thank the Reviewer for his/her useful suggestions. Citation of the suggested article has been made in line 42. […] in the case of invasive aspergillosis and non-albicans species particularly with Candida auris and Candida parapsilosis [9–13]
The following reference has been added:
- Codda, G.; Willison, E.; Magnasco, L.; Morici, P.; Giacobbe, D.R.; Mencacci, A.; Marini, D.; Mikulska, M.; Bassetti, M.; Marchese, A.; et al. In Vivo Evolution to Echinocandin Resistance and Increasing Clonal Heterogeneity in Candida Auris during a Difficult-to-Control Hospital Outbreak, Italy, 2019 to 2022. Euro Surveill 2023, 28, 2300161, doi:10.2807/1560-7917.ES.2023.28.14.2300161.
- A table reporting methods, vantage and disadvantages could represent a good way to summarize the review results.
Line 441. A list of advantages and disadvantages for all phenotypic fungicidal tests has been made in Table 1.
Table 1. List of advantages and disadvantages for all phenotypic fungicidal tests
|
Procedures |
Advantages |
Disadvantages |
|
Minimum Fungicidal Concentration |
Results interpretation;
Reduced costs |
Standardization of the procedure
Lack of clinical correlation |
|
Time-Kill Curve analysis |
Standardization of the procedure;
Results interpretation |
Technical expertise required, time-consuming
Lack of clinical correlation |
|
Serum Fungicidal Concentration |
Standardization of the procedure;
Reduced cost
Results with direct clinical correlations |
Technical expertise required;
Results interpretation
Requires extra blood samples to be taken from patients
Analyses to be performed in the shortest amount of time possible due to the reduced stability of the antimicrobial drugs after sampling. |
Comments on the Quality of English Language
Minor editing of the English language is required.
Reviewer 2 Report
Comments and Suggestions for Authors
In this review Franconi & Lupetti perform a substantive search of the literature to bring together data that goes beyond the standard MIC testing and how this can influence clinical decisions.
This review is timely and of scientific interest. I only have a few comments;
- This reviewer would prefer if the title does not state "yeasts" but "Candida". Most literature is from Candida field and very little is done to explain about other yeasts (a little bit of Cryptococcus but does not warrant this title).
- Most Candida species mentioned here have undergone a name change. This reviewer does not mind the old name but mention the new updated names where it is mentioned first.
- No mention is made about how most antifungals are fungistatic and therefore killing might not be a good measure. This needs to be stated and that for most clinical use fungistatic effects are sought.
- In the first mention of tolerance, it should be stated how the authors define tolerance. There are many different and contradictive definitions out there, especially when it comes to fungi. Try and link the data referred to, to this definition. Some papers have other definitions and therefore other outcomes.
- Might benefit also mentioning that persistence has been shown for A. fumigates recently by Scott et al 2023.
Author Response
Reviewer #2
In this review Franconi & Lupetti perform a substantive search of the literature to bring together data that goes beyond the standard MIC testing and how this can influence clinical decisions.
This review is timely and of scientific interest. I only have a few comments;
- This reviewer would prefer if the title does not state "yeasts" but "Candida". Most literature is from Candida field and very little is done to explain about other yeasts (a little bit of Cryptococcus but does not warrant this title).
We thank the reviewer for his/her kindful observation and knowledgeable suggestions. The title has been changed accordingly as follows:
In-vitro susceptibility tests in the context of Antifungal Resistance: beyond Minimum Inhibitory Concentration in Candida spp.
- Most Candida species mentioned here have undergone a name change. This reviewer does not mind the old name but mention the new updated names where it is mentioned first.
The following statements have been added:
Line 43. Candida glabrata (currently classified as Nakaseomyces glabrata)
Line 349. Candida krusei (currently renamed Pichia kudriavzevii)
- No mention is made about how most antifungals are fungistatic and therefore killing might not be a good measure. This needs to be stated and that for most clinical use fungistatic effects are sought.
Line 687. Apart from that, it also important to mention that killing effect can only be evaluated for fungicidal drugs. However, in clinical practice, most antifungal drugs used are fungistatic and therefore a restricted group of drugs and pathogens can be the subject of prospective fungicidal test studies.
- In the first mention of tolerance, it should be stated how the authors define tolerance. There are many different and contradictive definitions out there, especially when it comes to fungi. Try and link the data referred to, to this definition. Some papers have other definitions and therefore other outcomes.
Line 46. According to the definition proposed by Bearman and colleagues [25], antifungal tolerance is defined as the ability of a subpopulation of a susceptible isolate to grow over a longer incubation period (extended beyond the time required to define MIC values) under drug concentrations above MIC values, without harboring any known genetic resistance mechanism.
The reference for the definition is:
- Berman, J.; Krysan, D.J. Drug Resistance and Tolerance in Fungi. Nat Rev Microbiol 2020, 18, 319–331, doi:10.1038/s41579-019-0322-2.
- Might benefit also mentioning that persistence has been shown for A. fumigates recently by Scott et al 2023.
Line 464. Persistence has also been observed in A. fumigatus in relation to its fungicidal drug voriconazole [93]. Based on the findings reported in their study Scott et al. [93] suggest that persistence may play a potential role in treatment failure for voriconazole in A. fumigatus infections.
The following reference has been added:
- Scott, J.; Valero, C.; Mato-López, Á.; Donaldson, I.J.; Roldán, A.; Chown, H.; Van Rhijn, N.; Lobo-Vega, R.; Gago, S.; Furukawa, T.; et al. Aspergillus Fumigatus Can Display Persistence to the Fungicidal Drug Voriconazole. Microbiol Spectr 2023, 11, e0477022, doi:10.1128/spectrum.04770-22.
Reviewer 3 Report
Comments and Suggestions for Authors
Dear Authors
This article presents a very complete and interesting review of the different methods actually available to determine not only the appearing of resistant Candida spp. isolates by means of MIC tests but also the value of other available tests like MFC, SFC and tolerance to antifungals.
Anyway, I consider some of those tests are difficult to implement in many institutions.
Candida glabrata is nowadays included in genus Nakaseomyces (Nakaseomyces glabrata), please correct.
In line 156 you say: “…presented and MFC ≥2 x MIC, please correct
Author Response
Reviewer #3
Dear Authors
This article presents a very complete and interesting review of the different methods actually available to determine not only the appearing of resistant Candida spp. isolates by means of MIC tests but also the value of other available tests like MFC, SFC and tolerance to antifungals.
Anyway, I consider some of those tests are difficult to implement in many institutions.
Candida glabrata is nowadays included in genus Nakaseomyces (Nakaseomyces glabrata), please correct.
We thank Reviewer #3 for his/her kindful observations and suggestions. All the manuscript has been revised accordingly and correct names of isolates along with previous classification have been reported
Line 44. Candida glabrata (currently classified as Nakaseomyces glabrata)
Line 349. Candida krusei (currently renamed Pichia kudriavzevii)
In line 156 you say: “…presented and MFC ≥2 x MIC, please correct
Line 176. (76% of isolates presented MFC ≥2 x MIC)